# Early Sepsis Prediction Using Publicly Available Data: High-Performance AI/ML Models with First-Hour Clinical Information

**DOI:** 10.3390/diagnostics15212727

**Published:** 2025-10-28

**Authors:** Hao Wang, Destiny Pounds, Wenhui Zhang, Alaa Y. Mokbel, Md Niamul Kabir, Xin Yao Lin, April Highlander, Iman Dehzangi

**Affiliations:** 1Department of Emergency Medicine, JPS Health Network, 1500 S. Main St., Fort Worth, TX 76104, USA; hwang@ies.healthcare; 2Department of Biomedical Data Science, School of Applied Computational Sciences, Meharry Medical College, Nashville, TN 37203, USA; dpounds24@mmc.edu; 3Center for Data Science, Nell Hodgson Woodruff School of Nursing, Emory University, 1520 Clifton Rd, Atlanta, GA 30322, USA; wenhui.zhang@emory.edu; 4Department of Neuroscience, Georgetown University Medical Center, New Research Building, 3970 Reservoir Rd, NW, Washington, DC 20057, USA; aym16@georgetown.edu; 5Department of Natural Science, College of Arts & Science, Albany State University, 504 College Drive, Albany, GA 31705, USA; mdniamul.kabir@asurams.edu; 6Weill Cornell Medicine, 420 East 70th Street, Office LH-303, New York, NY 10021, USA; xyl4001@med.cornell.edu; 7Department of Psychiatry and Human Behavior, Alpert Medical School of Brown University, Providence, RI 02912, USA; april_highlander@brown.edu; 8Department of Computer Science, Center for Computational and Integrative Biology, Rutgers University, Camden, NJ 08102, USA; 9Rutgers Cancer Institute, Rutgers University, New Brunswick, NJ 08901, USA

**Keywords:** AI/ML, sepsis, early detection, algorithm

## Abstract

**Objectives:** Early identification of sepsis is critical, as delayed diagnosis significantly increases morbidity and mortality. We aimed to develop and validate AI/ML models for early sepsis prediction using structured electronic health record (EHR) data, waveform data, and a combination of both. **Methods:** We conducted a retrospective observational study using the AIM-AHEAD60 subset of the CHoRUS dataset. Adult patients (≥18 years) with a final diagnosis of sepsis were included. Structured EHR data (demographics, initial vital signs, laboratory results) and waveform data (continuous vital signs) from the first hour of hospital arrival were extracted. Three algorithms (i.e., XGBoost, LightGBM, and HistGB) were developed with a focus on maximizing the performance metric of recall. Other performance metrics were also assessed, including accuracy, precision, F1 score, and the area under the receiver operating characteristic curve (AUROC). **Results:** A total of 11,312 unique patients met the inclusion criteria, among whom 2245 individuals (19.85%) were diagnosed with sepsis at least once. Using structured EHR data alone, laboratory variables such as lactate and leukocyte count were most predictive. Waveform models identified respiratory rate, systolic blood pressure, and temperature trends in the first hour as key predictors. Combined models highlighted mean temperature and mean systolic blood pressure as top features. XGBoost achieved the highest AUROC (0.922) across all data configurations, with a recall above 80%, demonstrating robust performance despite substantial missing data. **Conclusions:** High-performing AI/ML models for early sepsis prediction can be developed from publicly available datasets using only first-hour clinical information. XGBoost models demonstrate strong potential for real-time clinical screening.

## 1. Introduction

Sepsis, defined as life-threatening organ dysfunction caused by a dysregulated host response to infection, remains a major global health concern despite advances in medical care [1]. It is associated with high morbidity, mortality, and healthcare costs. According to the Global Burden of Disease Study, sepsis affects an estimated 49 million people annually and contributes to 11 million deaths, representing nearly 20% of global mortality [2]. While the overall mortality rate of sepsis has declined in recent decades due to earlier recognition and timely intervention, it remains a leading cause of in-hospital death worldwide [3].

Historically, sepsis diagnosis relied on meeting at least two systemic inflammatory response syndrome (SIRS) criteria in conjunction with a suspected or confirmed source of infection [4]. However, SIRS is limited by low specificity and the inability to distinguish infectious from non-infectious inflammatory conditions [5]. To address these shortcomings, subsequent scoring systems have been developed. The Sequential Organ Failure Assessment (SOFA) score and its simplified quick SOFA (qSOFA) variant aim to quantify organ dysfunction and provide bedside screening, respectively [1]. SOFA is accurate in ICU settings but requires laboratory data, making it less practical for rapid triage, whereas qSOFA is quick but suffers from low sensitivity in early-stage sepsis [6]. Other scoring systems include the Modified Early Warning Score (MEWS), which is widely implemented for detecting clinical deterioration but is nonspecific to sepsis, and the National Early Warning Score (NEWS), which incorporates vital signs into a weighted system and has been adopted in some national health services [6,7,8]. Each scoring system offers trade-offs: those with broader applicability often lack specificity, and those with high accuracy may be limited to specialized settings or require laboratory inputs that delay detection.

Over the past decade, artificial intelligence (AI) and machine learning (ML) have emerged as promising approaches for early detection of sepsis. Numerous AI/ML models have been developed, particularly in ICU populations, leveraging electronic health record (EHR) data, including laboratory results, vital signs, imaging, and demographic variables [9,10,11]. For example, the InSight algorithm demonstrated AUROC values around 0.88 for predicting sepsis onset several hours before clinical recognition using only very limited clinical and vital sign inputs [11]. Deep learning approaches such as the AI Clinician model have also been explored to recommend optimal treatment strategies for sepsis [12]. Gradient boosting models and deep learning methods have shown better predictive performance using structured EHR data than the traditional sepsis scoring systems [13]. Despite encouraging results, most models are trained in ICU populations, which may limit generalizability, and many rely on continuous monitoring data or extensive laboratory results that are unavailable during initial patient evaluation. These factors constrain their deployment in emergency departments or low-resource settings, where early detection is equally critical.

Timely recognition is crucial because mortality risk increases with each hour of delay in initiating appropriate antimicrobial therapy. Kumar et al. reported a 7.6% increase in mortality for every hour antimicrobial administration was delayed after the onset of hypotension in septic shock [14]. Therefore, the ideal prediction model should detect sepsis within 1–2 h of hospital presentation, enabling prompt intervention and improving patient outcomes.

Publicly available clinical datasets have facilitated the development of reproducible and generalizable models for predicting sepsis. In addition to the widely used Medical Information Mart for Intensive Care (MIMIC-IV) dataset, other notable resources include the eICU Collaborative Research Database, containing multi-center ICU data from over 200 hospitals and the PhysioNet Challenge datasets, which have supported multiple international competitions on early sepsis prediction [15,16,17]. Studies using MIMIC-IV have reported AUROC values in the range of 0.80–0.90 for sepsis prediction, while models developed using eICU data have demonstrated external validity across diverse hospital settings [11,18]. The advantages of these datasets include open access, reproducibility, heterogeneous patient populations, and the ability to benchmark models across institutions. Therefore, it is crucial to utilize such datasets for the development of AI/ML models that predict sepsis within the first hours of presentation, as this aligns with the current emphasis on early detection and generalizability. However, using publicly available data to develop AI/ML models for very early sepsis prediction is still underexplored.

In this study, we aim to develop AI/ML models for early sepsis prediction using the publicly available CHoRUS dataset. Our goal is to predict sepsis within the first hour of hospital presentation using only basic demographic information, vital signs, and initial laboratory results. By focusing on readily available features, our approach seeks to produce a prediction tool that is both generalizable and feasible for use across diverse healthcare settings, including those with limited diagnostic resources.

## 2. Methods

### 2.1. Study Setting and Design

This retrospective observational study was conducted to develop AI/ML models for early sepsis prediction using the CHoRUS dataset (https://ca-aimahead-www.lemoncliff-697a56e6.eastus2.azurecontainerapps.io/ (accessed on 5 January 2025). We specifically utilized data from the AIM-AHEAD60 subset of CHoRUS, formatted according to the Observational Medical Outcomes Partnership (OMOP) Common Data Model. Because the dataset is fully de-identified and contains no personally identifiable health information (PHI), the study met criteria for exemption from institutional review board (IRB) oversight, and the requirement for informed consent was waived.

### 2.2. Inclusion and Exclusion Criteria

Eligible participants were adult patients (aged ≥18 years) with a final diagnosis of sepsis. Patients without the diagnosis of sepsis were used as the control group. We excluded patients without any documented diagnoses in the electronic health record (EHR) and all pediatric patients (<18 years).

### 2.3. Primary Outcome

The primary outcome was the presence of a final clinical diagnosis of sepsis, regardless of infection source or anatomical location.

### 2.4. Data Preprocessing

All available features were retained, even if they contain missing values. Extreme outliers were capped based on predefined thresholds to reduce the influence of biologically implausible values and potential data entry errors. Specifically, the following values were capped: Age, with values greater than 90 years capped at 90 to comply with public dataset de-identification protocols. Heart rate: Values > 250 beats per minute were capped at 250. Respiratory rate: Values > 40 breaths per minute were capped at 40. Systolic blood pressure: Values > 300 mmHg were capped at 300. Temperature: All temperatures were converted from Fahrenheit to Celsius; values < 25 °C were capped at 25 °C, and those >50 °C were capped at 50 °C.

### 2.5. Key Features

Two major categories of data were incorporated: (1) Structured EHR data, including demographics: age (continuous) and sex (male/female); Initial vital signs: heart rate, respiratory rate, systolic blood pressure, pulse oximetry, temperature; Laboratory results: lactate, band count, platelet count, creatinine, bilirubin, neutrophil count. Only values available within the first hour of patient arrival were included. For example, if a patient arrived at 08:00 and lactate results were available by 08:59, lactate was included; if results were reported at 09:01, the feature was excluded for that encounter. This approach inevitably resulted in missing data for many variables, but these were retained for model training. (2) Waveform data: continuous monitoring data for heart rate, respiratory rate, systolic blood pressure, and temperature. Data from the first hour of presentation were summarized by calculating the mean, minimum, and maximum values for each parameter. Finally, the structured and waveform datasets were combined for model development.

### 2.6. Model Development

The data was split into 70% for training and 30% for testing. Three AI/ML algorithms were evaluated: Extreme Gradient Boosting (XGBoost), Light Gradient Boosting Machine (LGBM), and Histogram-based Gradient Boosting (HistGB). Model development proceeded in four sequential stages: (1) baseline modeling using default algorithm parameters; (2) Class imbalance adjustment to address skewed sepsis prevalence. Given the lower prevalence of sepsis cases, we addressed the class imbalance using algorithm-specific weighting methods. For instance, in XGBoost, we applied the scale_pos_weight parameter to proportionally increase the weight of positive (sepsis) instances during training. This approach enabled the model to place greater emphasis on minority class examples without altering the underlying data distribution. (3) Hyperparameter tuning to optimize performance and minimize overfitting; and (4) Recall prioritization to maximize sensitivity for sepsis detection, thereby reducing the risk of missed high-risk cases. Final decision thresholds were calibrated to maximize recall (sensitivity), given the clinical imperative to identify all potential sepsis cases.

### 2.7. Performance Accuracy

Model performance was evaluated on both the training and testing datasets using multiple classification metrics, including accuracy, recall (also known as sensitivity), precision (also known as positive predictive value), F1 score, and the area under the receiver operating characteristic curve (AUROC).

### 2.8. Model Interpretability

To enhance clinical interpretability, feature importance scores were extracted from models that provide intrinsic importance measures. These were analyzed to identify the most influential predictors of sepsis onset.

### 2.9. Statistical Analysis

Hyperparameter tuning was performed separately for each algorithm (XGBoost, LightGBM, and HistGB) using a randomized search strategy to optimize model performance while mitigating overfitting. The parameter search space included learning rate, maximum tree depth, number of estimators, subsampling ratio, and column sampling ratio, among other model-specific parameters. A randomized search was implemented using five-fold stratified cross-validation on the training set to preserve the proportion of sepsis and non-sepsis cases within each fold.

Class imbalance in the training data was addressed using algorithm-specific weighting schemes based on the ratio of negative to positive cases in the training set. No synthetic oversampling or undersampling was applied to preserve the natural class distribution and prevent the introduction of synthetic data artifacts.

For structured EHR data, missingness was retained as an informative signal rather than imputed, given prior evidence that patterns of missingness may carry predictive value in clinical datasets. For waveform features, missing values in derived summary statistics (mean, minimum, maximum) were also retained without imputation. The final optimized models were evaluated on the held-out test set using the performance metrics described above. A summarized AI/ML model development pipeline diagram was shown in Figure 1. Patients were categorized into two groups based on sepsis status (sepsis vs. non-sepsis). Continuous variables were compared using Student’s *t*-test, while Pearson’s chi-square test was applied to assess differences in categorical variables between the groups. A two-sided *p*-value of <0.05 was considered statistically significant. All analyses were performed in Python (version 3.8).

## 3. Results

A total of 11,312 individuals were included in the final analysis; among all these individuals, 2245 (19.85%) patients were diagnosed with sepsis at least once. The general characteristics of the study population are summarized in Table 1. Briefly, 57.81% of the individuals in the entire study were male, 71.42% identified as White, and 6.10% were Hispanic. Individuals in the sepsis group appear to be slightly older than those in the non-sepsis group, although there are no statistically significant differences in terms of sex, race, and ethnicity between these two groups (Table 1 and Appendix A).

### 3.1. Identification of the Best AI/ML Model for Early Sepsis Prediction

We evaluated three distinct AI/ML algorithms to determine the optimal approach for early sepsis prediction in our study population. Each model was developed according to a predefined multi-step development process. The comparative performance metrics for all models are presented in Table 2. Among the algorithms tested, XGBoost consistently demonstrated the highest predictive accuracy, including recall and AUROC, across all input configurations, including structured EHR data alone, waveform data alone, and the combined dataset incorporating both sources. These results indicate that XGBoost provided the most robust and generalizable performance for early sepsis detection within our study cohort.

### 3.2. AI/ML Early Sepsis Prediction Model Interpretation

When using structured EHR data alone, laboratory findings, particularly lactate and leukocyte counts, emerged as the most influential predictors in the model. When waveform (vital signs) data were analyzed independently, respiratory rate, systolic blood pressure, and temperature measured toward the end of the first hour of monitoring contributed most strongly to prediction. In the combined dataset, mean temperature and mean systolic blood pressure were among the leading predictors for early sepsis detection. Collectively, these findings highlight the crucial role of vital signs, both as static measures and as evolving patterns over time, in enhancing the accuracy of early sepsis prediction (Figure 2).

Feature importance values are shown for models trained using (A) structured electronic health record (EHR) data, (B) waveform-flattened data, and (C) a combination of EHR and waveform data. The horizontal bars represent the relative contribution of each variable to the prediction of early sepsis, with longer bars indicating higher importance.

## 4. Discussion

In this study, we used a large, publicly available dataset to develop AI/ML models for early sepsis prediction. Our findings demonstrate that it is feasible to construct high-performing predictive models using open-access data sources, which may contain different formats, including structured EHR data, waveform data, or a combination of the two. Notably, our models achieved AUROC values of approximately 0.90 using either structured or waveform datasets independently, indicating a strong discriminative ability even when data inputs are restricted. Across all evaluated algorithms, XGBoost consistently produced the highest predictive performance, aligning with prior research that has identified gradient boosting-based methods as among the most effective for clinical prediction tasks, including sepsis detection [19,20].

The XGBoost model achieved an AUROC of 0.92, indicating excellent discriminatory ability to detect early sepsis cases. In addition to AUROC, we also assessed traditional metrics such as accuracy and F1 score (Table 2). While AUROC remains the most robust metric across varying decision thresholds, we observed that accuracy and F1-score varied across models, reflecting trade-offs between sensitivity and precision under class imbalance. The F1-score, in particular, provides a useful balance between recall and precision, and its moderate values highlight the clinical complexity of early sepsis identification. These metrics complement AUROC in providing a more nuanced understanding of the model’s practical performance.

A distinctive feature of our approach is its reliance on the limited health information available within the first hour of a patient’s arrival at the healthcare facility. This constraint reflects real-world clinical scenarios, particularly in emergency or low-resource settings where complete laboratory panels, imaging, or specialist evaluations may not be immediately available. Early triage using vital signs plays a critical role in identifying disease severity, particularly in the context of sepsis. By definition, sepsis involves a confirmed or suspected source of infection accompanied by systemic inflammatory response syndrome (SIRS). The SIRS criteria encompass abnormalities in temperature, heart rate, and respiratory rate, highlighting the importance of vital signs in initial risk stratification. When patients meet SIRS criteria, their risk of sepsis is significantly elevated. In clinical practice, lactate measurement is a commonly employed diagnostic tool when sepsis is suspected. A lactate level equal to or greater than 2 mmol/L is often considered indicative of early tissue hypoperfusion in the setting of infection and is associated with the worst outcomes [8]. Sensitivities for elevated lactate in sepsis range from 66% to 83%, with specificities ranging from 80% to 85% depending on the cutoff used (typically between 1.6 and 2.5 mmol/L) [8]. Additionally, a complete blood count (CBC) and a comprehensive metabolic panel (CMP) are routinely ordered in the ED. Leukocytosis (WBC > 12,000/μL) is both a common marker of infection and a component of the SIRS criteria. Thrombocytopenia, elevated creatinine (suggestive of renal dysfunction), and elevated bilirubin (indicating hepatic impairment) are all associated with progression to severe sepsis [21]. Importantly, these laboratory values are typically available within the first hour of ED presentation, supporting their feasibility for incorporation into early predictive models for sepsis identification.

Meanwhile, our study incorporated key sociodemographic variables based on established epidemiologic evidence linking these factors to sepsis risk and outcomes. Age is a well-documented predictor, with both the incidence and mortality of sepsis rising significantly among older adults, particularly those over 85 years [22]. Gender differences have also been consistently observed, with males generally exhibiting a higher risk of developing sepsis and experiencing worse clinical outcomes compared to females [23]. Race and ethnicity are important determinants of health disparities in sepsis. Prior research has shown that non-White populations, including Black and Hispanic individuals, are more likely to experience higher rates of sepsis-related hospitalization and mortality [24]. These disparities likely reflect a complex interplay of structural factors, including socioeconomic status, access to healthcare, and potential biological susceptibility. Marital status has emerged as another influential social determinant, with unmarried individuals demonstrating greater odds of sepsis-related hospitalization [25]. This may reflect the protective effects of social support, which can influence health-seeking behavior, medication adherence, and early detection of illness. Although the inclusion of sociodemographic factors in predictive modeling enhances the fairness and contextual relevance of early sepsis identification, their predictive utility reflects associative correlations rather than direct clinical causality. Our analysis indicates that these variables serve a complementary role, enriching the model’s interpretability and equity considerations, but do not serve as primary indicators. In contrast, early clinical indicators, such as vital signs and laboratory results, remain the most robust and immediate predictors for timely sepsis detection.

Moreover, we intentionally built and tested our models on datasets with a substantial proportion of missing values, mimicking the incomplete data environment often encountered in practice. The ability of XGBoost to handle missing data natively was particularly advantageous, supporting its robustness and practical applicability in frontline clinical decision support systems [26,27]. When model interpretation was performed using only structured EHR data, the top three predictive features for early sepsis were lactate, leukocyte count, and body temperature. These variables are widely recognized as key clinical indicators of infection and systemic inflammation, aligning closely with established sepsis diagnostic criteria and current emergency care practices. In contrast, when models were trained on waveform data, the most influential predictors were derived from later-recorded vital signs, highlighting the value of continuous clinical monitoring. This suggests that sepsis may not be fully apparent within the first hour of presentation to the ED. During this early period, treatment may not yet be initiated, while the underlying pathophysiological process of sepsis continues to evolve. As a result, vital signs recorded later during the ED stay may better reflect the disease trajectory and thus carry greater predictive weight than initial readings. When structured EHR and waveform data were combined, the model identified a more complex and nuanced set of predictive features, incorporating a mixture of vital signs and laboratory values, which underscored the multifactorial nature of sepsis risk. These findings emphasize that effective early sepsis prediction models must account for both timing and diversity of clinical data inputs to capture the dynamic presentation of sepsis.

Our results also reaffirm previous studies, which report that XGBoost outperforms many alternative algorithms for sepsis prediction [19,28,29]. Its strengths include efficient handling of heterogeneous feature types, robustness to multicollinearity, and the capacity to learn nonlinear relationships between predictors and outcomes. Importantly, in the context of early sepsis prediction, these capabilities enable meaningful predictions using both static demographic/laboratory data, as well as dynamic physiological measurements. Our deliberate exclusion of time-intensive variables, such as imaging findings, reflects a design choice aimed at maximizing early applicability and minimizing prediction delays.

From a clinical standpoint, AI/ML models can only be adopted if their performance metrics align with real-world priorities. In sepsis care, missing a diagnosis may lead to rapid deterioration, multi-organ failure, and death, whereas false positives, while potentially leading to unnecessary work-up, are far less likely to result in severe harm [30]. Consequently, in our modeling, we prioritized maximizing recall (sensitivity) over precision, aiming to minimize false negatives even at the expense of increased false positives. This trade-off aligns with established sepsis management guidelines, which emphasize early and aggressive intervention when sepsis is suspected.

This study offers several notable strengths. First, we utilized a large-scale, publicly available dataset derived from 15 sites nationwide, thereby increasing the likelihood that our findings are applicable across diverse healthcare settings. Second, by incorporating both structured EHR and waveform data, we demonstrated the feasibility of developing robust models from diverse data sources, thereby broadening their potential applicability. Third, our emphasis on the first hour of presentation, combined with tolerance for substantial missing data, mirrors real-world emergency care conditions and enhances translational potential.

## 5. Limitations

Our work also has limitations. First, as a retrospective analysis, it is subject to the inherent biases and data quality issues of observational datasets, including missing, incomplete, or erroneous records, especially regarding individuals’ sociodemographic characteristics. For example, there was a significant amount of missing “marital status” information, resulting in less value for this variable in model prediction. Second, although we compared three commonly used algorithms, other advanced models, such as deep learning architectures, hybrid models, or ensemble frameworks, may offer superior performance and warrant future evaluation. Third, we focused only on a set of well-established predictors from prior sepsis literature; incorporating additional domains, such as psychosocial determinants or pre-hospital care behaviors, could further improve early detection. Fourth, only structured and waveform data were used; unstructured data (e.g., clinical notes) and streaming real-time signals were not considered. Fifth, though our model predicts sepsis well, we did not compare the performance accuracy of sepsis prediction between our model and other well-established models in the literature (such as SOFA, qSOFA). Finally, while class weighting addressed outcome-level imbalance, we acknowledge demographic imbalances, particularly in ethnic representation. We did not perform extensive explainability analysis beyond feature importance, nor did we evaluate model fairness across demographic subgroups, which are critical considerations for clinical deployment.

## 6. Future Directions

Future research should explore a broader range of public datasets, integrate unstructured and multimodal inputs, and evaluate a wider spectrum of AI/ML algorithms, including deep neural networks and ensemble learning methods. Additionally, thorough model interpretability assessments and fairness audits are essential before clinical implementation to ensure transparency, trust, and equitable performance across populations. External validation in prospective cohorts, ideally in real-time clinical workflows, will also be necessary to confirm performance in operational environments.

## 7. Conclusions

Our findings demonstrate that robust AI/ML models for early sepsis prediction can be developed using publicly available datasets, with limited first-hour clinical and laboratory data, including vital signs, lactate, CBC, and CMP. XGBoost achieved the best performance across structured, waveform, and combined data, with high sensitivity, making it well-suited for clinical screening. These results highlight the potential for generalizable, rapid, and practical sepsis prediction tools, warranting further validation and expansion to additional data types and real-time settings.

## Figures and Tables

**Figure 1 diagnostics-15-02727-f001:**
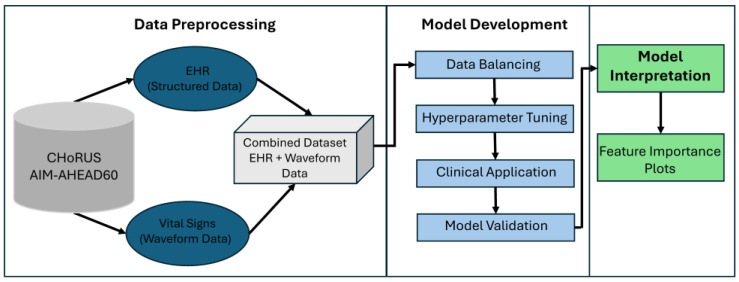
AI/ML Model Development Pipeline Diagram.

**Figure 2 diagnostics-15-02727-f002:**
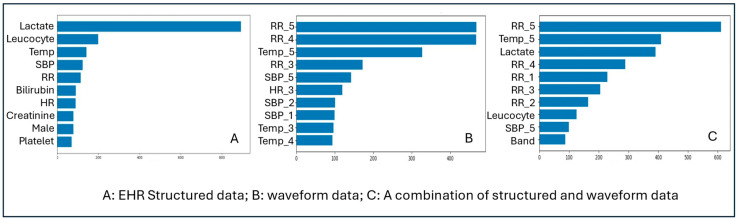
Feature Importance of XGBoost Prediction of Early Sepsis Using Different Datasets.

**Table 1 diagnostics-15-02727-t001:** General Characteristics of the Study Population.

	Sepsis Group (n = 2245)	Non-Sepsis Group (n = 9067)	*p*
Age—years Mean (SD) Median (IQR)	62 (17) 65 [53–75]	61 (17) 64 [51–74]	0.0129 0.0145
Sex—n (row percentage%) Male Female Others	1322(20.22) 923 (19.34) 0 (0)	5217 (79.78) 3849 (80.66) 1 (100)	0.4548
Race—n (row percentage%) White Black Asian Others	1596 (19.75) 266 (19.22) 81 (21.83) 302 (20.43)	6483 (80.25) 1118 (80.78) 290 (78.17) 1176 (79.57)	0.6539
Ethnicity—n (row percentage%) Hispanic/Latino Non-Hispanic/Latino Others	133 (19.28) 1950 (20.07) 162 (17.86)	557 (80.72) 7765 (79.93) 745 (82.14)	0.2593
Marital Status—n (row percentage%) Partnered/Married Divorced/Separated Single Widowed Missing *	58 (35.58) 16 (43.24) 32 (26.67) 13 (39.39) 2126 (19.40)	105 (64.42) 21 (56.76) 88 (73.33) 20 (60.61) 8833 (80.60)	<0.001

* A missing value indicates that marital status was not recorded in the dataset. “[]” means the IQR (interquartile range).

**Table 2 diagnostics-15-02727-t002:** Performance Accuracy of Different AI/ML models for Early Sepsis Predictions.

	XGBoost		LightGBM		HistGB	
	Training	Testing	Training	Testing	Training	Testing
EHR Structured Data (demographics and laboratory tests)						
Accuracy	0.739	0.738	0.986	0.968	0.977	0.970
Precision	0.093	0.086	0.771	0.399	0.586	0.428
Recall	0.967	0.901	0.714	0.351	0.522	0.370
F1 score	0.169	0.156	0.741	0.374	0.552	0.397
AUROC	0.951	0.913	0.977	0.911	0.942	0.914
Waveform Data						
Accuracy	0.836	0.826	0.883	0.883	0.932	0.920
Precision	0.196	0.173	0.212	0.211	0.366	0.287
Recall	0.833	0.740	0.575	0.571	0.657	0.502
F1 score	0.318	0.281	0.310	0.308	0.470	0.365
AUROC	0.915	0.871	0.857	0.851	0.931	0.875
A Combination of EHR structured and Waveform Data						
Accuracy	0.868	0.858	0.915	0.913	0.945	0.934
Precision	0.245	0.218	0.290	0.278	0.441	0.363
Recall	0.905	0.811	0.586	0.564	0.722	0.571
F1 score	0.386	0.344	0.388	0.372	0.547	0.443
AUROC	0.953	0.922	0.906	0.895	0.957	0.920

Abbreviations: XGBoost, extreme gradient boosting; LightGBM, Light Gradient Boosting Machine; HistGB, Histogram-based Gradient Boosting; AUROC, area under the receiver operating characteristic curve.

## Data Availability

CHoRUS AIM-AHEAD60 will become publicly available database shortly. By the time of this submission, these data are not ready to be publicly released. Data can be obtained upon request to the corresponding author.

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
