# Peer review of "Early Sepsis Prediction Using Publicly Available Data: High-Performance AI/ML Models with First-Hour Clinical Information"

_diagnostics, 2025, doi:10.3390/diagnostics15212727_

Round 1

Reviewer 1 Report

Comments and Suggestions for Authors

This paper is dedicated to a very serious and important problem of early prediction of sepsis. The objective was to predict sepsis at least within the first hour upon patient’s admission to the hospital.

The authors used XGBoost, LightGBM, and HistGB ML models. A reasonably large dataset of 11,312 individuals was selected from publicly available sources. The study was conducted for different demographic groups. It implies that the authors expected to find some differences in early sepsis predictions for different demographic groups.

Comment 1: There is a conflict between information in Table 1 and the description in lines 194-195 where proportion between White and Hispanic groups is presented. Table 1 shows Whites as a race and Hispanic as ethnicity. Why is focus on Hispanic ethnicity is of interest for the research while other ethnicities are shown as “others”?

Comment 2: One may assume that marital status may impact on sepsis, but then there are many similar parameters, maybe patient’s height and weight, profession, and many other parameters. Why did the authors choose marital status as one of the main parameters?

Comment 3: There is ethnic proportion of 71.42% of White and 6.10% of Hispanic patients in the dataset. Was it caused just by fact or were there some other considerations for this proportion?

Comment 4: 57.81% of individuals were males. It presumes that the rest were females. What was the reason for this uneven proportion with females?

Comment 5: The authors claim that the dataset was balanced. Different recall and precision (Table 2) may indicate unbalanced dataset, though their difference does not directly imply it. However, a significant difference in ethnic proportion (see Comment 3) may indicate imbalance if ethnicity was a parameter for analysis. Anyway, it would be helpful to describe how the dataset was balanced.

Comment 6: The problem statement of this research was formulated in lines 106-108 as “In this study, we aim to develop AI/ML models for early sepsis prediction using the 106 publicly available CHoRUS dataset. Our goal is to predict sepsis within the first hour of 107 hospital presentation using only basic demographic information, vital signs, and initial 108 laboratory results.” However, the answer to this question was not explicitly discussed in the paper, neither in the Discussion section (lines 233 …) nor in the Conclusion (line 298 …). Instead, most discussion was around the comparison of the three engaged ML models.

Comment 7: The authors introduced many parameters lime ethnicity, gender, even marital status but had never discussed their impact on sepsis in the results of their study, except that ethnicity does not have

Comment 8: Figure 2 should be improved because all text in the figure is too small and cannot be read.

Comment 9: Figure 2 should be discussed in the text otherwise its meaning is not clear.

Comment 10: Section “Conclusion” is supposed to wrap up on the paper and discuss major outcomes. However, this section is very short and ranks the ML models used in the research. Is this paper about a benchmark of ML models or about early sepsis prediction?

Author Response

Manuscript ID: diagnostics-3894180

Title: “Early Sepsis Prediction Using Publicly Available Data: High-Performance AI/ML Models with First-Hour Clinical Information”

To: Editorial Board

Dear Editor,

Thank you for allowing a resubmission of our manuscript with an opportunity to address the Reviewer’s comments.

We would like to express our sincere gratitude to the Reviewers for their thoughtful and constructive feedback. Their comments have been invaluable in helping us improve the quality of our manuscript. We have carefully considered each point raised and have made revisions accordingly. Below, we provide a detailed response to each comment, highlighting the changes made and the rationale behind them. We believe these revisions have strengthened the paper, and we hope the updated version meets the Reviewers’ expectations.

We are uploading (a) our point-by-point response to the comments (below) (response to Reviewers), (b) and an updated manuscript highlighting the changes in blue color.

Best regards,

Authors

Reviewer #1

Comment 1: There is a conflict between information in Table 1 and the description in lines 194-195 where proportion between White and Hispanic groups is presented. Table 1 shows Whites as a race and Hispanic as ethnicity. Why is focus on Hispanic ethnicity is of interest for the research while other ethnicities are shown as “others”?

Thanks to the Reviewer for highlighting this point and apologies for the confusion. The description in lines 194-195 is “column percentages”. For example: “57.81% individuals were male, 71.42% identified as White, and 6.10% were Hispanic.” This indicates that 57.81% of the total male individuals were among the entire study cohort. However, in Table 1, we used the “row percentage,” which indicates the percentage of each item between the “sepsis” and “non-sepsis” groups. For example, there were 1596 (19.75%) White individuals in the sepsis group in comparison to 6483 (80.25%) of White individuals in the non-sepsis group. In terms of the focus on Hispanic ethnicity, whereas other ethnicities are shown as “others”, it is mainly due to two reasons: 1) these variables were harvested from the existing dataset, which is hard to change; 2) Hispanic ethnicity in general is grounded in both public health relevance and population-specific disparities. Hispanic individuals represent one of the fastest-growing ethnic groups in the U.S. and are frequently identified as being at higher risk for certain health disparities, including limited access to care and underdiagnosis. To clarify, we have revised our manuscript (see Lines 202-203 and revised Table 1).

Comment 2: One may assume that marital status may impact on sepsis, but then there are many similar parameters, maybe patient’s height and weight, profession, and many other parameters. Why did the authors choose marital status as one of the main parameters?

Thanks to the Reviewer for the valuable comments. Marital status was selected as a key variable based on existing evidence that social determinants of health, particularly social support structures, can significantly influence outcomes in critical illness, including sepsis. Numerous studies have shown that married individuals often experience better health outcomes due to increased emotional, logistical, and caregiving support. In the context of sepsis, this support may facilitate earlier recognition of symptoms, timely medical attention, better adherence to treatment, and post-discharge care coordination, which can directly impact survival and recovery. While it is true that many other patient-level variables, such as height, weight, or occupation, may influence sepsis outcomes, the inclusion of every possible parameter is not always feasible or clinically justified, especially when such data are unavailable or inconsistently recorded. As you can see here in Table 1, even with the “marital status” variable, there is so much missing information that makes this variable less valuable for model prediction. We addressed these issues in our limitations section (see Lines 337-339).

Comment 3: There is ethnic proportion of 71.42% of White and 6.10% of Hispanic patients in the dataset. Was it caused just by fact or were there some other considerations for this proportion?

The dataset we used primarily collected data from 15 sites nationwide within the US territory. We believe this is caused just by the fact.

Comment 4: 57.81% of individuals were males. It presumes that the rest were females. What was the reason for this uneven proportion with females?

The dataset we used primarily collected data from 15 sites nationwide within the US territory. We believe this is caused just by the fact.

Comment 5: The authors claim that the dataset was balanced. Different recall and precision (Table 2) may indicate unbalanced dataset, though their difference does not directly imply it. However, a significant difference in ethnic proportion (see Comment 3) may indicate imbalance if ethnicity was a parameter for analysis. Anyway, it would be helpful to describe how the dataset was balanced.

We appreciate the reviewer’s insightful observation regarding the implications of differing recall and precision values, as well as the potential imbalance across ethnic subgroups. When we refer to the dataset as “balanced,” we are specifically referencing the distribution of the outcome variable (i.e., the classes used in model training). Prior to model development, we examined the class distribution and applied appropriate resampling techniques to address any class imbalances that were present. For example, in our method section (Line 159), it was addressed: “2) Class imbalance adjustment to address skewed sepsis prevalence”. We ensured that both positive and negative outcome classes were equally represented using the hyperparameter tuning, such as the “positive weight class” in XGBoost. Regarding ethnic representation and imbalance, we acknowledge that the ethnic distribution in the dataset was not uniform, as noted in Comment 3. However, ethnicity was not used as a direct predictor in the final modeling framework; therefore, its imbalance does not affect the performance of the core outcome classification model. Different values of precision and recall, particularly when recall is higher, can reflect not only class imbalance but also model sensitivity or emphasis placed during optimization. In our case, the model was tuned to prioritize recall (i.e., minimizing false negatives, see Lines 164-168) due to the clinical relevance of identifying all true cases, even if that resulted in a lower precision. This decision aligns with established clinical risk trade-offs and was not influenced by a significant class imbalance after preprocessing.

To address the Reviewer’s recommendation, we have now added a detailed description of how class balancing was performed in the Methods section (see Lines 159-164), including the exact steps and rationale. We’ve also noted limitations regarding subgroup imbalances (e.g., ethnicity) and how future studies could explore stratified modeling or weighted fairness-aware approaches (see Lines 349-353).

Comment 6: The problem statement of this research was formulated in lines 106-108 as “In this study, we aim to develop AI/ML models for early sepsis prediction using the 106 publicly available CHoRUS dataset. Our goal is to predict sepsis within the first hour of 107 hospital presentation using only basic demographic information, vital signs, and initial 108 laboratory results.” However, the answer to this question was not explicitly discussed in the paper, neither in the Discussion section (lines 233 …) nor in the Conclusion (line 298 …). Instead, most discussion was around the comparison of the three engaged ML models.

We thank the Reviewer’s valued comments. We revised our manuscript to include an additional paragraph in the discussion section, specifically addressing the main objectives of this study and highlighting the importance of collecting first-hour clinical and laboratory data. Further discussed the clinical features of these first-hour collected variables. We also expand our conclusion to include a detailed list of first-hour clinical and laboratory features in our study. (see Discussion and Conclusion section, Lines 254-271, and Lines 364-365)

Comment 7: The authors introduced many parameters lime ethnicity, gender, even marital status but had never discussed their impact on sepsis in the results of their study, except that ethnicity does not have

Thanks to the Reviewer for highlighting this point. We revised our manuscript to include an additional paragraph discussing the sociodemographic characteristics associated with sepsis (see Discussion section, Lines 272-288).

Comment 8: Figure 2 should be improved because all text in the figure is too small and cannot be read.

 Thanks to the Reviewer for highlighting this point.  To address this comment, we revised our Figure 2.

Comment 9: Figure 2 should be discussed in the text otherwise its meaning is not clear.

Yes, we revised the Discussion section to include more discussion of Figure 2 (see Lines 293-309).

Comment 10: Section “Conclusion” is supposed to wrap up on the paper and discuss major outcomes. However, this section is very short and ranks the ML models used in the research. Is this paper about a benchmark of ML models or about early sepsis prediction?

Thanks to the Reviewer for highlighting this point. We revised our conclusion section to emphasize the early sepsis prediction (See Lines 364-365).

Reviewer 2 Report

Comments and Suggestions for Authors

This retrospective observational study was conducted to develop AI/ML models for early sepsis prediction using the publicly available CHoRUS dataset. The work presented for review concerns an important topic. The research is interesting and presented correctly. I have few comments.

  1. Provide a table with summarized data (Structured EHR and Waveform data) for Sepsis and Non-Sepsis groups
  2. It would be interesting to compare the accuracy of your AI/ML models for early sepsis prediction with SIRS, SOFA, and qSOFA.
  3. "In this study, we aim to develop AI/ML models for early sepsis prediction using the publicly available CHoRUS dataset" (line 106) - Add a link to the dataset, please
  4. The section “Discussion” is quite brief. It would be helpful to describe in more detail.
  5. Please increase the font size in the Figure 2.

Author Response

Manuscript ID: diagnostics-3894180

Title: “Early Sepsis Prediction Using Publicly Available Data: High-Performance AI/ML Models with First-Hour Clinical Information”

To: Editorial Board

Dear Editor,

Thank you for allowing a resubmission of our manuscript with an opportunity to address the Reviewer’s comments.

We would like to express our sincere gratitude to the Reviewers for their thoughtful and constructive feedback. Their comments have been invaluable in helping us improve the quality of our manuscript. We have carefully considered each point raised and have made revisions accordingly. Below, we provide a detailed response to each comment, highlighting the changes made and the rationale behind them. We believe these revisions have strengthened the paper, and we hope the updated version meets the Reviewers’ expectations.

We are uploading (a) our point-by-point response to the comments (below) (response to Reviewers), (b) and an updated manuscript highlighting the changes in blue color.

Best regards,

Authors.

Reviewer #2

  1. Provide a table with summarized data (Structured EHR and Waveform data) for Sepsis and Non-Sepsis groups

 Thanks to the Reviewer for highlighting this point. To address the Reviewer’s comment, we provided a supplemental Table to include summarized data for the sepsis and non-sepsis groups.

  1. It would be interesting to compare the accuracy of your AI/ML models for early sepsis prediction with SIRS, SOFA, and qSOFA.

We thank the Reviewer for the insightful comment. We referenced the SIRS criteria in our discussion because SIRS-related variables (e.g., temperature, heart rate, respiratory rate, and WBC count) were included in our prediction model. Consequently, the model's performance may exceed that of SIRS-based assessment alone, particularly in identifying early sepsis risk. However, we acknowledge the limitations inherent in our retrospective dataset. Specifically, the dataset lacked sufficient granularity to reliably assess mental status, which is a key component of both the SOFA and qSOFA scores. As a result, we were unable to compute SOFA or qSOFA for comparison. We have revised the manuscript to reflect this limitation explicitly in the Limitations section (see Lines 346349).

  1. "In this study, we aim to develop AI/ML models for early sepsis prediction using the publicly available CHoRUS dataset" (line 106) - Add a link to the dataset, please

 Thanks to the Reviewer for highlighting this point. We have revised our manuscript to include the link (see Lines 115-116).

  1. The section “Discussion” is quite brief. It would be helpful to describe in more detail.

 Thanks to the Reviewer for highlighting this point. We revised and significantly expanded our discussion section  (see revised Discussion section).

  1. Please increase the font size in the Figure 2.

Yes, revised.

Round 2

Reviewer 1 Report

Comments and Suggestions for Authors

1) The paper is very confusing. For example, demographic profile may be adequate for statistical analysis and general healthcare review. However, once the patient is admitted, demography does not play any role in one-hour early sepsis diagnostics. At that time, only clinical data play the decisive role.

2) The choice of clinical parameters for early sepsis detection is quite shallow. It looks like demographic data play the more important role than clinical parameters.

3) Diagnostic accuracy AUROC metric is quite adequate. However, the author presented tradictiona accuracy and F1 score in Table 2, but have never discussed them later in the paper. Section "Discussion" just mentioned AUROC = 0.9 and ignored the spread in accuracy and F1 scores presented in Table 2.

4) Section "Conclusion" actually contains no conclusions on early sepsis prediction. Thus, it is not clear about the results and value of this research in early sepsis prediction. 

5) The paper mostly discusses technological (XGBoost, LightGBM, and HistGB) rather than sepsis detection itself. It looks like the paper is mostly focused on technology rather than on early sepsis detection problem.

6) The logical flow of the paper is quite unfocused, and it leaves a quite confusing impression. The paper should clearly state the research problems and purpose and then logically follow along that line. Conclusions should contain the answers to the research questions stated in the research problem.

7) I do not believe the paper can be published as is or after a slight revision. It should be fundamentally rewritten before its resubmission.

Author Response

Reviewer #1

  • The paper is very confusing. For example, demographic profile may be adequate for statistical analysis and general healthcare review. However, once the patient is admitted, demography does not play any role in one-hour early sepsis diagnostics. At that time, only clinical data play the decisive role.

Response: We thank the reviewer for the thoughtful comment. While we understand the concern, we would like to emphasize that demographic characteristics, such as age, gender, and race, have been shown in numerous peer-reviewed studies to significantly contribute to sepsis prediction models. In our study, these factors complemented key clinical indicators, such as lactate, leukocyte count, and temperature, which were among the top predictors in our model (see Figure 2). Moreover, because our objective was to develop a model for early sepsis prediction within the first hour of patient presentation, detailed clinical parameters may not yet be fully available, and most of these patients had not yet been admitted. Therefore, both clinical and demographic variables were carefully selected based on their immediate availability and predictive value in the earliest phase of care.

See the following:

Demographic characteristics—including age, sex, race/ethnicity, and socioeconomic status—play critical roles in sepsis prediction, influencing both incidence and outcomes. Older age is a strong risk factor for sepsis, with incidence and mortality rising sharply in the elderly; younger children have a high risk due to immune immaturity.[1-3] Male sex is associated with higher sepsis rates and mortality compared to females.[2][4] Racial and ethnic disparities are evident: Black and certain minority populations experience higher incidence and worse outcomes, partly due to differences in comorbidities, access to care, and socioeconomic factors.[2][4-6] Socioeconomic deprivation, functional limitations, and not being in the labor force further increase sepsis risk and mortality.[5][7-8] Incorporating demographic and social determinants into predictive models improves accuracy beyond clinical scores alone, underscoring their importance in risk stratification and outcome prediction.[8-10] In summary, demographic characteristics are essential for accurate sepsis prediction and should be integrated into clinical risk models and epidemiologic surveillance.

  1. Meyer NJ, Prescott HC. Sepsis and Septic Shock. The New England Journal of Medicine. 2024;391(22):2133-2146. doi:10.1056/NEJMra2403213.
  2. Arina P, Hofmaenner DA, Singer M. Definition and Epidemiology of Sepsis. Seminars in Respiratory and Critical Care Medicine. 2024;45(4):461-468. doi:10.1055/s-0044-1787990.
  3. Henriksen DP, Pottegård A, Laursen CB, et al. Risk Factors for Hospitalization Due to Community-Acquired Sepsis - A Population-Based Case-Control Study. PloS One. 2015;10(4):e0124838. doi:10.1371/journal.pone.0124838.
  4. Angus DC, van der Poll T. Severe Sepsis and Septic Shock. The New England Journal of Medicine. 2013;369(9):840-51. doi:10.1056/NEJMra1208623.
  5. Hennessy DA, Soo A, Niven DJ, et al. Socio-Demographic Characteristics Associated With Hospitalization for Sepsis Among Adults in Canada: A Census-Linked Cohort Study. Canadian Journal of Anaesthesia = Journal Canadien d'Anesthesie. 2020;67(4):408-420. doi:10.1007/s12630-019-01536-z.
  6. Schertz AR, Lenoir KM, Bertoni AG, et al. Sepsis Prediction Model for Determining Sepsis vs SIRS, qSOFA, and SOFA. JAMA Network Open. 2023;6(8):e2329729. doi:10.1001/jamanetworkopen.2023.29729.
  7. van Staa TP, Pate A, Martin GP, et al. Sepsis and Case Fatality Rates and Associations With Deprivation, Ethnicity, and Clinical Characteristics: Population-Based Case-Control Study With Linked Primary Care and Hospital Data in England. Infection. 2024;52(4):1469-1479. doi:10.1007/s15010-024-02235-8.
  8. Sarraf E, Sadr AV, Abedi V, Bonavia AS. Enhancing Sepsis Prognosis: Integrating Social Determinants and Demographic Variables Into a Comprehensive Model for Critically Ill Patients. Journal of Critical Care. 2024;83:154857. doi:10.1016/j.jcrc.2024.154857.
  9. Montull B, Menéndez R, Torres A, et al. Predictors of Severe Sepsis Among Patients Hospitalized for Community-Acquired Pneumonia. PloS One. 2016;11(1):e0145929. doi:10.1371/journal.pone.0145929.
  10. Seymour CW, Kennedy JN, Wang S, et al. Derivation, Validation, and Potential Treatment Implications of Novel Clinical Phenotypes for Sepsis. JAMA. 2019;321(20):2003-2017. doi:10.1001/jama.2019.5791.

  • The choice of clinical parameters for early sepsis detection is quite shallow. It looks like demographic data play the more important role than clinical parameters.

Response: We appreciate the reviewer’s thoughtful feedback and the opportunity to clarify our approach. Our model was intentionally designed to integrate both clinical and demographic variables to reflect the multifactorial nature of early sepsis detection. As demonstrated in Figure 2, the top three predictors: lactate, leukocyte count, and temperature, are all clinically relevant parameters that directly reflect underlying infection and systemic inflammatory responses.

These clinical markers were prioritized based on extensive prior literature supporting their early prognostic value in sepsis and their practical feasibility for rapid assessment during the initial hour of patient presentation. While demographic factors were also included to enhance the model’s generalizability across diverse patient populations, clinical features were clearly the most influential drivers of prediction in our model. By emphasizing readily available, time-sensitive clinical indicators alongside relevant demographic characteristics, our model aims to provide actionable insights. We believe this balanced approach aligns with clinical priorities and enhances both the validity and utility of our prediction framework.

  • Diagnostic accuracy AUROC metric is quite adequate. However, the author presented tradictiona accuracy and F1 score in Table 2, but have never discussed them later in the paper. Section "Discussion" just mentioned AUROC = 0.9 and ignored the spread in accuracy and F1 scores presented in Table 2.

Response: We thank the reviewer for this insightful comment. To improve clarity and completeness, we have now incorporated a brief discussion of the accuracy and F1 score metrics into the revised Discussion section (see lines 255-263). While AUROC was our primary metric due to its threshold-independent nature and wide use in binary classification, we recognize that accuracy and F1-score provide additional insights, particularly in the context of class imbalance and predictive reliability. Specifically, we acknowledge the observed variation in accuracy and F1 scores across different models in Table 2 and have now added a sentence to contextualize these values in relation to AUROC and clinical applicability. This clarification allows for a more balanced interpretation of overall model performance.

  • Section "Conclusion" actually contains no conclusions on early sepsis prediction. Thus, it is not clear about the results and value of this research in early sepsis prediction. 

Response: We appreciate the reviewer’s thoughtful comment. As noted in the Conclusion section of our manuscript, “Our findings demonstrate that robust AI/ML models for early sepsis prediction can be developed using publicly available datasets, with limited first-hour clinical data.” This clearly reflects our focus on early sepsis prediction, which was a central objective of the study. Our model was intentionally designed to rely on clinical and demographic features that are typically available within the first hour of a patient’s arrival. This constraint was both methodologically intentional and clinically motivated, aiming to ensure the real-world applicability of our approach for early risk stratification and prompt intervention. We have clarified this focus throughout the manuscript, particularly in the Introduction and Conclusion, to highlight that our predictive model was developed using limited, early-available data.

  • The paper mostly discusses technological (XGBoost, LightGBM, and HistGB) rather than sepsis detection itself. It looks like the paper is mostly focused on technology rather than on early sepsis detection problem.

Response: We thank the reviewer for this insightful comment. In our revised manuscript, we discussed intensely not only the technology but also the clinical significance of early sepsis detection.

  • The logical flow of the paper is quite unfocused, and it leaves a quite confusing impression. The paper should clearly state the research problems and purpose and then logically follow along that line. Conclusions should contain the answers to the research questions stated in the research problem.

Response: We sincerely thank the reviewer for the thoughtful feedback. The primary objective of our study is to develop and evaluate artificial intelligence/machine learning (AI/ML) models for early sepsis detection, which is explicitly stated in the “Objectives” section (Lines 106–111, final paragraph of the Introduction). To support this aim, we implemented a modeling framework that incorporated three distinct data modalities: (1) structured electronic health record (EHR) data alone; (2) high-resolution waveform data alone; and (3) a combined EHR and waveform dataset. This multimodal approach allowed us to systematically assess the incremental contribution of each data type to predictive performance. As such, our conclusions were purposefully aligned with the study’s design to affirm the feasibility and clinical utility of using AI/ML models to support sepsis detection during the critical early window of patient care. To ensure clarity and consistency throughout the manuscript, we have carefully reviewed and confirmed that the objectives, methods, results, and conclusions remain well-aligned. We greatly appreciate the reviewer’s insights, which helped us refine our message and enhance the clarity of our manuscript.

  • I do not believe the paper can be published as is or after a slight revision. It should be fundamentally rewritten before its resubmission.

Response: We have revised our manuscript extensively.

Reviewer 2 Report

Comments and Suggestions for Authors

Dear Authors,

Thank you for your detailed responses to the comments.

- Please specify the units of measurement in the Supplementary Table.

- In the Statistical Analysis section, specify which method was used for group differences (Table 1 and Suppl.Table).

Author Response

Reviewer#2

Dear Authors,

Thank you for your detailed responses to the comments.

- Please specify the units of measurement in the Supplementary Table.

Yes, it is revised.

- In the Statistical Analysis section, specify which method was used for group differences (Table 1 and Suppl.Table).

Yes, we revised with the inclusion of methods used to compare groups in the Methods section (see Lines 196-199)

Round 3

Reviewer 1 Report

Comments and Suggestions for Authors

I appreciate the authors for providing the detailed answers and revising the paper.

There is no doubt that demographic parameters such as age, gender, and race may play an important role in early sepsis prediction due to physiological and biochemical differences they imply. I have only one concern left which is related to the impact of marital and socioeconomic status of the admitted patients on short-term (one-hour) early sepsis prediction. The statistical data on these parameters in the datasets show their impact on the risk of sepsis (in the future) among general population. However, the sample of the already admitted patients is fundamentally different from the sample from general population. Marital and socio-economic status impacts on long-term risk of sepsis. The risk factors in the sample from general population have already been realized once the patients were admitted to the hospital. There would be higher proportion of the people with higher risk of sepsis in the sample of the admitted patients.

There are two questions for the authors:

  • If the authors believe that the marital and socio-economic status significantly impact on the short-term (one hour) sepsis prediction, they have to justify it.
  • (Lines 286-296 in the revised manuscript): “Prior research has shown that non-White populations, including Black and Hispanic individuals, are more likely to experience higher rates of sepsis-related hospitalization and mortality.[25] These disparities likely reflect a complex interplay of structural factors, including socioeconomic status, access to healthcare, and potential biological susceptibility. Marital status has emerged as another influential social determinant, with unmarried individuals demon-strating greater odds of sepsis-related hospitalization.[26] This may reflect the protective effects of social support, which can influence health-seeking behavior, medication adherence, and early detection of illness.” The above statement has grounds and is supported by the references provided. However, the next sentence in the same paragraph: “Taken together, the inclusion of these sociodemographic factors in predictive modeling is both evidence-based and essential for improving the accuracy and equity of early sepsis identification strategies” cannot be logically derived by the preceding statement due to significant differences in two samples, one from general population and another from the admitted patients.

Early sepsis prediction is a very important problem, and clear solution will save many lives. It would be great if the authors clarify the concerns above before the paper is accepted for publication.

Author Response

There is no doubt that demographic parameters such as age, gender, and race may play an important role in early sepsis prediction due to physiological and biochemical differences they imply. I have only one concern left which is related to the impact of marital and socioeconomic status of the admitted patients on short-term (one-hour) early sepsis prediction. The statistical data on these parameters in the datasets show their impact on the risk of sepsis (in the future) among general population. However, the sample of the already admitted patients is fundamentally different from the sample from general population. Marital and socio-economic status impacts on long-term risk of sepsis. The risk factors in the sample from general population have already been realized once the patients were admitted to the hospital. There would be higher proportion of the people with higher risk of sepsis in the sample of the admitted patients.

There are two questions for the authors:

  • If the authors believe that the marital and socio-economic status significantly impact on the short-term (one hour) sepsis prediction, they have to justify it.

Response: We appreciate the reviewer’s thoughtful comments regarding the relevance of marital and socioeconomic status (SES) in the context of early sepsis prediction within the first hour of hospital admission. While it is true that marital and SES factors are traditionally associated with long-term health outcomes and disease susceptibility in the general population, there is growing evidence that these variables can also contribute to short-term clinical risk stratification, even in acutely ill, hospitalized populations. We include these variables in our model for the following reasons: 1) indirect Impact on baseline health status: marital status and SES are often linked to baseline health, access to care, and medication adherence. These factors may shape the severity of illness at presentation, potentially influencing key clinical parameters (e.g., delayed care-seeking behavior, under-treatment of chronic conditions) observable upon arrival; 2) influence on health seeking behavior and timing of ED Arrival: for example, individuals with lower SES or those who are unmarried may delay seeking care due to transportation issues, cost concerns, or lack of social support, resulting in more advanced clinical presentations of sepsis upon ED arrival. Such a delay can affect vital signs, laboratory markers, and early clinical impressions captured in the first hour, thereby indirectly influencing prediction accuracy; 3) use in model calibration rather than causal Interpretation: we do not interpret SES and marital status as causal drivers of sepsis within one hour. Rather, their inclusion helps improve model calibration and generalizability by accounting for heterogeneity in the population that could influence the availability, timing, or severity of observable clinical parameters; and 4) previous reviews in predictive modeling: prior studies in sepsis predictive modeling literature have used similar demographic and socioeconomic indicators to enhance prediction accuracy (see the following references)

We agree with the reviewer that the role of marital and SES factors in short-term sepsis onset is likely limited. Their predictive value arises from associative patterns, not direct clinical causality. In addition, our predictive model also revealed that marital factors are not the top features that contribute to the model prediction. To address this, we have clarified this point in the revised manuscript (see Discussion section, Lines 299-305).

“Although the inclusion of sociodemographic factors in predictive modeling enhances the fairness and contextual relevance of early sepsis identification, their predictive utility reflects associative correlations rather than direct clinical causality. Our analysis indicates that these variables serve a complementary role, enriching the model's interpretability and equity considerations, but do not serve as primary indicators. In contrast, early clinical indicators, such as vital signs and laboratory results, remain the most robust and immediate predictors for timely sepsis detection.”

References:

Sarraf E, Sadr AV, Abedi V, Bonavia AS. Enhancing Sepsis prognosis: Integrating social determinants and demographic variables into a comprehensive model for critically ill patients. J Crit Care. 2024 Oct:83:154857. doi: 10.1016/j.jcrc.2024.154857. Epub 2024 Jul 11.

Hauschildt K, Pan A, Bernstein T, et al. Consideration of Sociodemographics in Machine Learning-Driven Sepsis Risk Prediction. Crit Care Med. 2025 Sep 1;53(9):e1815-e1820.  doi: 10.1097/CCM.0000000000006741. Epub 2025 Jun 9.

  • (Lines 286-296 in the revised manuscript): “Prior research has shown that non-White populations, including Black and Hispanic individuals, are more likely to experience higher rates of sepsis-related hospitalization and mortality.[25] These disparities likely reflect a complex interplay of structural factors, including socioeconomic status, access to healthcare, and potential biological susceptibility. Marital status has emerged as another influential social determinant, with unmarried individuals demon-strating greater odds of sepsis-related hospitalization.[26] This may reflect the protective effects of social support, which can influence health-seeking behavior, medication adherence, and early detection of illness.” The above statement has grounds and is supported by the references provided. However, the next sentence in the same paragraph: “Taken together, the inclusion of these sociodemographic factors in predictive modeling is both evidence-based and essential for improving the accuracy and equity of early sepsis identification strategies” cannot be logically derived by the preceding statement due to significant differences in two samples, one from general population and another from the admitted patients.

Response: We sincerely thank the reviewer for this insightful observation. We agree that sociodemographic characteristics such as marital status and socioeconomic factors may not exert the same immediate influence on short-term, early sepsis prediction as clinical parameters available within the first hour of admission. While prior research supports the relevance of these factors in shaping long-term sepsis risk, our findings indicate that their predictive contribution is secondary compared to vital signs and laboratory indicators obtained at presentation. Accordingly, we have revised the sentence to clarify this point as follows (see Lines 299-305): 

“Although the inclusion of sociodemographic factors in predictive modeling enhances the fairness and contextual relevance of early sepsis identification, their predictive utility reflects associative correlations rather than direct clinical causality. Our analysis indicates that these variables serve a complementary role, enriching the model's interpretability and equity considerations, but do not serve as primary indicators. In contrast, early clinical indicators, such as vital signs and laboratory results, remain the most robust and immediate predictors for timely sepsis detection.”

Early sepsis prediction is a very important problem, and clear solution will save many lives. It would be great if the authors clarify the concerns above before the paper is accepted for publication.